# *Brd2* haploinsufficiency extends lifespan and healthspan in C57B6/J mice

Shilpa Pathak[1], William C. L. Stewart[1,2,3], Christin E. Burd[4,5], Mark E. Hester[6], David A. Greenberg[1,2]*

**1** Battelle Center for Mathematical Medicine, The Abigail Wexner Research Institute, Nationwide Children's Hospital, Columbus, Ohio, United States of America, **2** Department of Pediatrics, The Ohio State University, Columbus, Ohio, United States of America, **3** Department of Statistics, The Ohio State University, Columbus, Ohio, United States of America, **4** Department of Molecular Genetics, The Ohio State University, Columbus, Ohio, United States of America, **5** Department of Molecular Biology and Cancer Genetics, The Ohio State University, Columbus, Ohio, United States of America, **6** Steve and Cindy Rasmussen Institute for Genomic Medicine, Abigail Wexner Research Institute at Nationwide Children's Hospital, Columbus, Ohio, United States of America

* David.Greenberg@NationwideChildrens.org

**Data Availability Statement:** All relevant data are within the manuscript and Supporting Information files.

**Funding:** The authors received no specific funding for this work.

## Abstract

Aging in mammals is the gradual decline of an organism's physical, mental, and physiological capacity. Aging leads to increased risk for disease and eventually to death. Here, we show that *Brd2* haploinsufficiency (*Brd2*+/-) extends lifespan and increases healthspan in C57B6/J mice. In *Brd2*+/- mice, longevity is increased by 23% ($p$<0.0001), and, relative to wildtype animals (*Brd2*+/+), cancer incidence is reduced by 43% ($p$<0.001). In addition, relative to age-matched wildtype mice, *Brd2* heterozygotes show healthier aging including: improved grooming, extended period of fertility, and lack of age-related decline in kidney function and morphology. Our data support a role for haploinsufficiency of *Brd2* in promoting healthy aging. We hypothesize that Brd2 affects aging by protecting against the accumulation of molecular and cellular damage. Given the recent advances in the development of BET inhibitors, our research provides impetus to test drugs that target BRD2 as a way to understand and treat/prevent age-related diseases.

## Introduction

Inherent in the aging process is a gradual decline in physical, cognitive, and physiological capacity, an increasing risk of disease, and ultimately death. Although it is thought that aging results from the cumulative effects of molecular and cellular damage, we serendipitously discovered that a *Brd2*-haploinsufficient (*Brd2*+/-; denoted HET) mouse model we developed to study epilepsy [1–3] had a much longer lifespan compared to wild type (*Brd2*+/+; denoted WT) mice. In pursuing the mechanism by which BRD2, a bromodomain (BET) protein, predisposed to epilepsy [4, 5], we found that HETs, which are overtly normal [1, 3], not only have significantly longer lifespans but also show healthier-aging phenotypes, including reduced cancer incidence and improved kidney function, as compared to wildtype mice.

**Competing interests:** The authors have declared that no competing interests exist.

To date, there are only a handful of candidate longevity genes for C57B6/J mice. Of particular interest is *Cisd2*, which is perhaps the most well-known longevity gene in C57B6/J mice [6, 7]. *Cisd2*, which is chiefly concerned with the maintenance of mitochondrial integrity, increases lifespan in C57B6/J mice when overexpressed. The *Cisd2*-/- mouse (which is viable) has a shortened lifespan [7].

## BRD2 and longevity

There are several genes and molecular processes that are known to influence longevity in mice. Many of those genes are in turn influenced by Brd2. For example, *Brd2* haploinsufficiency downregulates IGF signaling [8], and IGF signaling is decreased in calorically restricted mice—a dietary intervention that increases lifespan [9, 10]. Similarly, *Brd2* haploinsufficiency up-regulates genes in the Sirtuin pathway [11], and up-regulation of the Sirtuin pathway is associated with increased lifespan [12, 13]. Specifically, Sirtuin 1 (SIRT1) and its homologs regulate longevity-related processes such as DNA repair, genome stability, inflammation, apoptosis, cell cycle progression and mitochondrial respiration [14–16]. Reduced expression of Brd2 also increases p53, Nqo1, and Hmox1 expression [11], all of which reduce oxidative stress. In addition, upregulation of p53 increases genomic stability, promotes DNA repair, and increases lifespan [17, 18]. Because *Brd2* haploinsufficiency is tied to multiple longevity-related genes and molecular processes, reduced expression of Brd2 could be a fundamental—and *heritable*—factor influencing lifespan.

## BRD2 and cancer

Whereas decreased Brd2 expression is associated with longevity-promoting processes, *increased* Brd2 expression can promote cancer in murine hematopoietic cells and in B-lymphocytes [19]. Furthermore, work from The Cancer Genome Atlas (TCGA) shows that *BRD2* expression is elevated across 32 distinct tumor types and establishes BRD2 as a promising drug target for human cancers. Also, reducing the expression of BRD2 in HeLa cells leads to a 60% increase in tumor-suppressing *p53* levels [20], which supports the notion that *increased BRD2* promotes cancer growth. Hence, the overexpression of BRD2 is oncogenic, whereas inhibiting the activity of BRD2 limits cancer progression. Furthermore, the overexpression of BET proteins in general promotes cancer in mice [19, 21, 22], and reduced BET expression (via BET inhibitors) is currently being tested as a treatment for cancer in both pre-clinical models and clinical trials [23–25]. Because cancer is a major cause of age-related morbidity and mortality, we hypothesize that Brd2's reduced expression could also increase healthspan by reducing cancer risk.

This report is the first demonstration that reducing Brd2 at the genetic level in a whole animal produces the same effects described above. We describe: 1) notable differences between Brd2 heterozygous and wildtype mice in aging-related phenotypes, including cancer incidence, kidney function, lifespan, and other aging-related measures, and, 2) evidence supporting the hypothesis that *Brd2*-haploinsufficiency increases lifespan and healthspan, in part, through the upregulation of cytoprotective genes.

## Materials and methods

All animal procedures were reviewed and approved under IACUC protocol (AR12-00016) at Nationwide Children's Hospital, Columbus, OH.

### Mouse model development and husbandry

*Brd2*+/- mice that express half the normal amount of *Brd2* [3] were generated according to previously described methods. Briefly, we performed targeted mutagenesis in SV129

embryonic stem cells (obtained from Baygenomics). Specifically, we used a Brd2 mutant embryonic stem cell line (RRE050) containing a gene trap vector pGT01 inserted into the first intron after the translational start site. This insertion abolished the epression of endogenous *Brd2*. We created *Brd2*+/- heterozygotes that carried one normal *Brd2* allele (+) and one loss-of-function *Brd2* allele (-). (A total loss of *Brd2* is embryonic lethal [3, 26]. These animals were then backcrossed to wild-type C57BL/6J mice for at least 10 generations to ensure a uniform genetic background. The mice were housed in a pathogen-free barrier environment for the duration of the study. Mice were kept at 72°F, with a 12/12-h light/dark cycle and had free access to water and rodent standard chow, containing 18.4% protein and 9.5% lipid. Heterozygous mice showed no difference in fecundity as compared to wildtype animals. The research team evaluated mice at least once a week, monitored the survival of each mouse daily. Additional clinical signs were monitored weekly including body weight, general body condition, pattern of respiration, dehydration, posture, movement, response to stimuli, and condition of coat hair. The husbandry staff monitored all mice during routine maintenance/health checks and contacted the attending veterinarian as needed. Trained husbandry personnel detected signs of illness/moribundness in mice (as noted above). Sick animals were immediately brought to the attention of the veterinarian and the research staff, who then evaluated whether the mice should be euthanized according to institutional animal care and use committee (IACUC) procedures. All animal procedures were reviewed and approved under our IACUC protocol (AR12-00016). Dates of birth and death were recorded, irrespective of whether an animal was found dead or euthanized.

## Mouse phenotyping at time of necropsy

All deceased mice were carefully screened for tumors (e.g. head and neck, liver, and kidney) and rectal prolapse at time of necropsy. The presence/absence of cataract was noted. Inflammation-related pathologies were determined at necropsy by inspection including swollen spleen, alopecia, skin lesions (e.g. dermatitis), and hardened kidney capsules. Inflammation in the kidney and other renal pathologies were later confirmed with microscopy. Specifically, formalin-fixed, paraffin embedded kidney samples from $Brd2^{+/-}$ and wildtype mice were stained using routine hematoxylin and eosin, Periodic acid-Schiff (PAS), or trichrome staining procedures prior to histological analyses. Histopathology of liver, spleen, and testes was analysed using hematoxylin and eosin staining.

## Mouse epigenetic aging clock

Genomic DNA was isolated from liver tissue samples from WT and HET mice using Quick DNA universal kit (Zymo Research). Evaluation of the biological age of these samples was done using the Mouse DNAge Epigenetic Aging Clock service offered by Zymo Research. This is a mouse biological age predictor based on DNA methylation levels of a small set of CpG sites [27]. This method is based on Hovrath's aging clock [28] and uses the Simplified Whole-panel Amplification Reaction Method.

## Nqo1, Hmox1 and Sirt1 mRNA expression

Given that Brd2 regulates cytoprotective genes and pathways, we looked for differential gene expression in selected cytoprotective genes. Total RNA was extracted from kidney tissue using an RNeasy mini kit (Qiagen), per manufacturer's instructions. Extracted RNA was then treated with DNAse I (Turbo DNA free, Invitrogen). Total RNA was reverse transcribed using iScript Reverse Transcription Supermix (Bio-Rad). Quantitative RT-PCR was performed using an Eppendorf thermocycler with TaqMan Gene Expression master mix (Applied Biosystems).

**Table 1. Genes analyzed by quantitative RT-PCR.**

| Gene | IDT assay ID |
|------|--------------|
| Nqo1 | Mm.PT.58.10871473 |
| Hmox1 | Mm.PT.58.8600055 |
| Sirt1 | Mm.PT.58.7263242 |
| Gapdh | Mm.PT.39a.1 |

Relative expression of the target genes was calculated with respect to a housekeeping gene (i.e. mGapdh). Assay identification numbers are listed in Table 1.

## p53 immunoblotting

p53 protein expression is strongly associated with tumor suppression [29], and we see reduced tumor incidence in our HET mice relative to wildtype. This observation led us to measure p53 expression in liver. Protein was extracted from the liver tissue of *Brd2* wildtype and heterozygous mice using RIPA buffer (Pierce) and total protein was quantified using BCA protein assays (Pierce). The protein samples were resolved on Mini-protean TGX stain-free gels (Bio-Rad) and transferred to a PVDF membrane. Primary antibodies directed against wild type p53 (Millipore Sigma, 1:1000) and Gapdh (loading control, Calbiochem, 1: 10,000) were used to probe the PVDF membrane and were detected by appropriate alkaline phosphatase-coupled antibodies (Santa Cruz Biotechnology). The resulting immunoblots were developed on a Typhoon FLA 9500 (GE Healthcare) imager after the addition of ECF substrate (GE Healthcare).

## Statistical procedures used in the analyses

To test differences in survival between WT and HET mice, we compared standard Kaplan-Meier estimates of the survivor functions using log-rank tests. To test whether the correlation between Brd2 and longevity depends on the presence or absence of tumor, we implemented a standard bootstrap procedure within groups (wildtype with tumors: WT+, wildtype without tumors: WT-, heterozygotes with tumors: HET+, heterozygotes without tumors: HET-). Specifically, for each bootstrap resample we computed the difference in median survival between WT+ and HET+, and the difference in median survival between WT- and HET-. Then, we constructed a confidence interval for the change in the difference of medians to test the null hypothesis of independence. Lastly, t-tests were used to compare differences in gene expression between WT and HET animals.

## Results

### Brd2-haploinsufficient mice show increased longevity

In our colony of *Brd2* heterozygous and wildtype mice, *Brd2*+/- mice showed a significant lifespan extension as compared to WT controls (Fig 1A). The median survival of HET mice was 954.5 days, ~ 23% longer than median survival of wildtype animals (775 days; $p < 0.0001$). The longest-lived mouse was a *Brd2* heterozygous animal that survived 1,528 days, and notably, this mouse had no tumors at necropsy. Moreover, the extended lifespan of HET mice was not affected by the presence or absence of tumors at necropsy ($p = 0.6$), i.e., overall, HET mice with tumors lived as long as HET mice without. However, there is suggestive evidence that sex influenced the degree to which lifespan increased in *Brd2*+/- mice compared to wildtype, (Fig 1B and 1C; bootstrap analysis; $p = 0.068$). Specifically, HET females showed a 46%

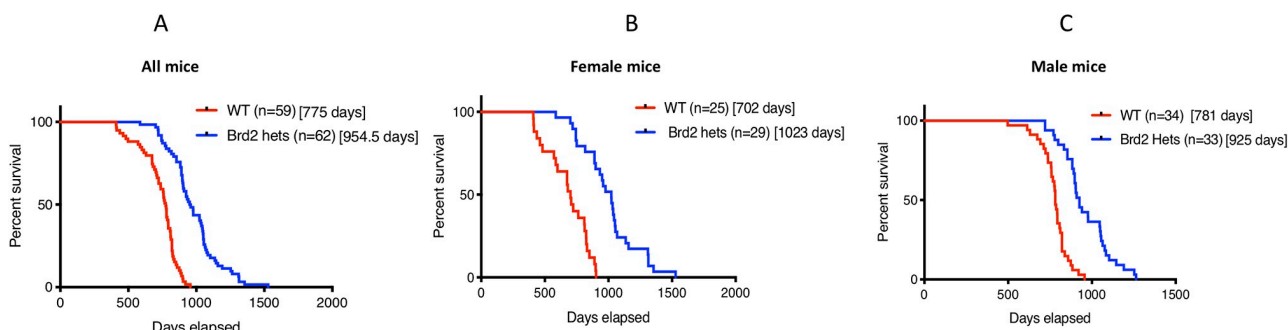

**Fig 1. Life span analysis of Brd2 HET mice.** Brd2 haploinsufficiency prolongs the life span in our mice colony (A). Panels B and C show the increase survival rates of both female (B) and male (C) mice separately (p<0.0001 for all groups). The figures in the square bracket indicate median lifespan.

extension in median life span compared to WT females (1,023 vs. 702 days). In contrast, HET males showed a 19% extension in median lifespan (925 vs 781 days).

## Brd2-haploinsufficient mice are biologically younger

We used Horvath's epigenetic clock (*aka* the Zymo clock) to determine the biological ages of 2 WT and 2 HET mice from biopsied liver samples; all 4 mice were age-matched (770 days). HET mice showed significantly younger liver tissue than WT (Table 1; *p*<0.012, SEM(WT): 4.5 and SEM(HET): 7.1). The biological age of the liver samples from the HET mice suggests that HET livers are less than half the biological age of WT livers. This is corroborated by liver histology, which reveals the loss of cytoarchitecture and inflammation in WT but maintenance of liver health histologically in HETs (S3 Fig). These data provide further confirmation that *Brd2* haplo-insufficiency increases lifespan and healthspan, and that it does so at a fundamental biological level.

## Brd2-haploinsufficient mice exhibit delayed aging

We confirmed haploinsufficiency of Brd2 transcripts by qPCR in a broad spectrum of tissues (S1 Fig).

Necropsy data suggested that *Brd2* heterozygous animals have delayed onset of aging. Because this mouse strain typically show signs of aging around 500 days, we examined the overt health of wildtype and heterozygous mice at this timepoint. At about 500 days, wildtype animals began to show symptoms of aging: reduced and slower movement, an "unkempt" appearance, hunching posture, declining health activity over the course of weeks, exhibiting an increasing decline in body condition, and slowly becoming hunched and lethargic. In contrast, HETs at the same age were youthfully active and remained healthy and fit in appearance until within about a week before death. These observations suggest that relative to WT, long-lived HET mice manifest age-related changes at a much older age and for a shorter period of time before death.

We also observed skin lesions in WT mice such as alopecia, dermatitis, and scarring at about 500 days. Age-matched HET mice showed sleek fur and no lesions. On necropsy, the elderly WT mice also had enlarged spleens (indicating chronic infection), an increased incidence of neoplastic lesions, and rectal prolapse compared to age-matched HET mice (Table 2). Next, we totaled the presence/absence of all 5 factors noted in Table 2: Dermatitis, Nephropathy, Rectal Prolapse, Lethargy and Moribund, and Ocular Lesions, (Table 2) and used this as a surrogate measure of health. Then, we performed logistic regression to test this measure for association with genotype while accounting for differences in lifespan. Based on this analysis,

**Table 2. Estimated biological age as determined by DNAge methylation clock.**

|  | Chronological age (Days) | Biological age (DNAge) (Days) |
|---|---|---|
| WT Female | 770 | 595 |
| WT Male | 770 | 532 |
| Het Female | 770 | 147.7 |
| Het Male | 770 | 247.1 |

it is clear that HET mice have fewer pathologies by the time of death than WT mice and the difference is highly statistically significant (p<0.00011).

Interestingly, the weights of the HETs and WT mice were comparable throughout their lives (see S2 Fig).

Pathologies present in the wildtype mice included: tumor, inflammation (including dermatitis), ocular lesions (e.g cataract, tumors, etc.), and nephropathy, which are all typical of the C57BL/6 strain (Table 3) [30]. By contrast, overt morbidities were reduced in *Brd2* heterozygous animals, with the most marked decreases being in tumor incidence and nephropathology (Table 3).

## Fertility differences between HET and WT

Preliminary experiments examining breeding performance of our HET males indicates that HET males remain fertile longer than WT males. In a preliminary experiment involving 8 month old males, we mated 6 WT and 6 HET males to normal cycling WT females. Of the 12 WT females mated to the 6 (8 month old) WT males only 1 female was impregnated, whereas, of the 12 WT females mated to the 6 (8 month old) HET males, all 12 females were impregnated (Fisher's Exact Test: $p$ = 0.015). Furthermore, we mated the same 6 WT males (above) to the 12 WT females originally impregnated by the 6 HET males (ie, proven fertile WT females); and none of the 12 WT females were impregnated by the WT males, providing added confirmation that at 8 month old WT males are no longer fertile but same-aged HET males are fertile.

Overall, the above observations on health status and fecundity demonstrate that, in addition to having a longer lifespan than WT mice, HET mice also have increased healthspan.

## HET mice have a lower incidence of cancer at necropsy

We observed a significant reduction in solid-tumor incidence in HET mice (13 out of 62) as compared to WTs (38 out of 59) for any solid tumor ($p$ = 0.0000014); this reduction was independent of sex (Table 3). The spectrum of types of cancer did not differ between HET and WT mice and included tumors associated with the liver and gastrointenstinal tract, as well as head

**Table 3. Major contributing causes of death in mice evaluated at end of life.**

|  | WT | HET |
|---|---|---|
|  | n (%) | n (%) |
| **Tumors** | 38/59 (64.4) | 13/62 (20.96) |
| **Dermatitis** | 25/59 (42.37) | 10/62 (16.12) |
| **Nephropathy** | 32/59 (54.23) | 9/62 (14.51) |
| **Rectal prolapse** | 14/59 (23.72) | 10/62 (16.12) |
| **Lethargy and moribund** | 28/59 (47.45) | 12/62 (19.35) |
| **Ocular lesions** | 18/59 (30.5) | 15/62 (24.19) |
| **No Apparent Pathology** | 19/59 (32.2) | 36/62 (58.06) |

and neck tumors. Among the mice with grossly visible neoplastic disease at necropsy, HET mice with visible solid tumors (median survival 925 days) showed longer lifespan than WT mice with visible solid tumors (median survival 781 days). The reduced tumor burden at death in the HET mice suggests that they are tumor-resistant as compared to WT mice. There was reduced cancer incidence in HET mice despite the increased time for tumor growth afforded by the 23% increase in lifespan. Moreover, the effect of Brd2 haploinsufficiency in extending lifespan in HET mice is independent of the presence or absence of tumors, since HET mice with tumors lived as long as HET mice without.

## Kidney pathology and function in HETs and WTs

Because kidney pathology is an age-related pathology in both mice and humans, we evaluated age-related histological changes to the kidney. We observed a much lower incidence of nephropathies in HET mice than WT (Table 2). WT mice at 500 days showed the expected age-appropriate renal pathologies [31] including elevated inflammatory cell foci (often lymphocytes and neutrophils) (*p = 0.0001*) and tubular cytoplasmic vacuolation (Fig 2A). We observed glomerulonephropathy with hyaline material expanding the mesangium (PAS positive), dilation of renal tubules and proteinaceous casts in aged WT (Fig 2C) mice. Additionally, levels of renal fibrosis were markedly increased in aged WT mice (Fig 2E). Not only were these pathologies not seen in age-matched HET mice (Fig 2B, 2D and 2F), but HETs also maintain healthier kidney function than their age-matched WT littermates. Specifically, blood urea nitrogen and creatinine levels of HET mice were lower than that of WTs (Fig 3A and 3B).

We chose to focus on kidney-related pathologies for two reasons. First, because the protective effect of *Brd2* haploinsufficiency was most pronounced in kidney (Table 3), and second, because the expression of cytoprotective genes in kidney allowed us to test our hypothesis that upregulation of cytoprotective genes leads to increased lifespan in HETs. However, in addition to the aforementioned kidney-related pathologies seen in WT mice, we also observed notable pathologies in other organs of the WT mice not seen in the HET mice. These included hyperplasia and cysts in liver, lymphoid hyperplasia in spleen, and testicular degeneration (S3 Fig).

## Upregulation of cytoprotective genes in Brd2 HET kidneys

It is well-recognized that the kidney aging process is accompanied by increased oxidative stress [32]. To test the effect of cytoprotective genes in HETs versus WTs, we evaluated expression of four cytoprotective genes in kidney: *heme oxygenase (HO)-1*, *NADPH quinone oxidoreductase 1* (*Nqo1*), Sirt1, and p53. We selected these genes because they are reported to be, in part, regulated by Brd2 [11, 20, 33]. The steady state mRNA levels of HO-1 and Nqo1, which are antioxidant proteins, were higher in HET kidneys as compared to WT kidneys. This suggests that the increased protection of kidney structure and function in HETs arises from the upregulation of their antioxidant defense system (Fig 3C and 3D). Sirt1, which also has a reno-protective effect [34], showed increased mRNA levels in HET kidneys as compared to WT kidneys (Fig 3E). Evaluation of p53, another cytoprotective gene, showed higher protein levels in HET kidneys compared to WT kidneys (Fig 3F). These observations support the hypothesis that Brd2 haploinsufficiency may influence longevity through the upregulation of cytoprotective genes.

## Discussion

Our *Brd2* haploinsufficient mouse model provides experimental evidence that reduced expression of *Brd2*—a gene closely tied to DNA, histones, chromatin structure, transcription, splicing, and possibly DNA methylation—can significantly alter the aging process, provide resistance to tumorigenesis, and delay the development of age-related pathologies in the

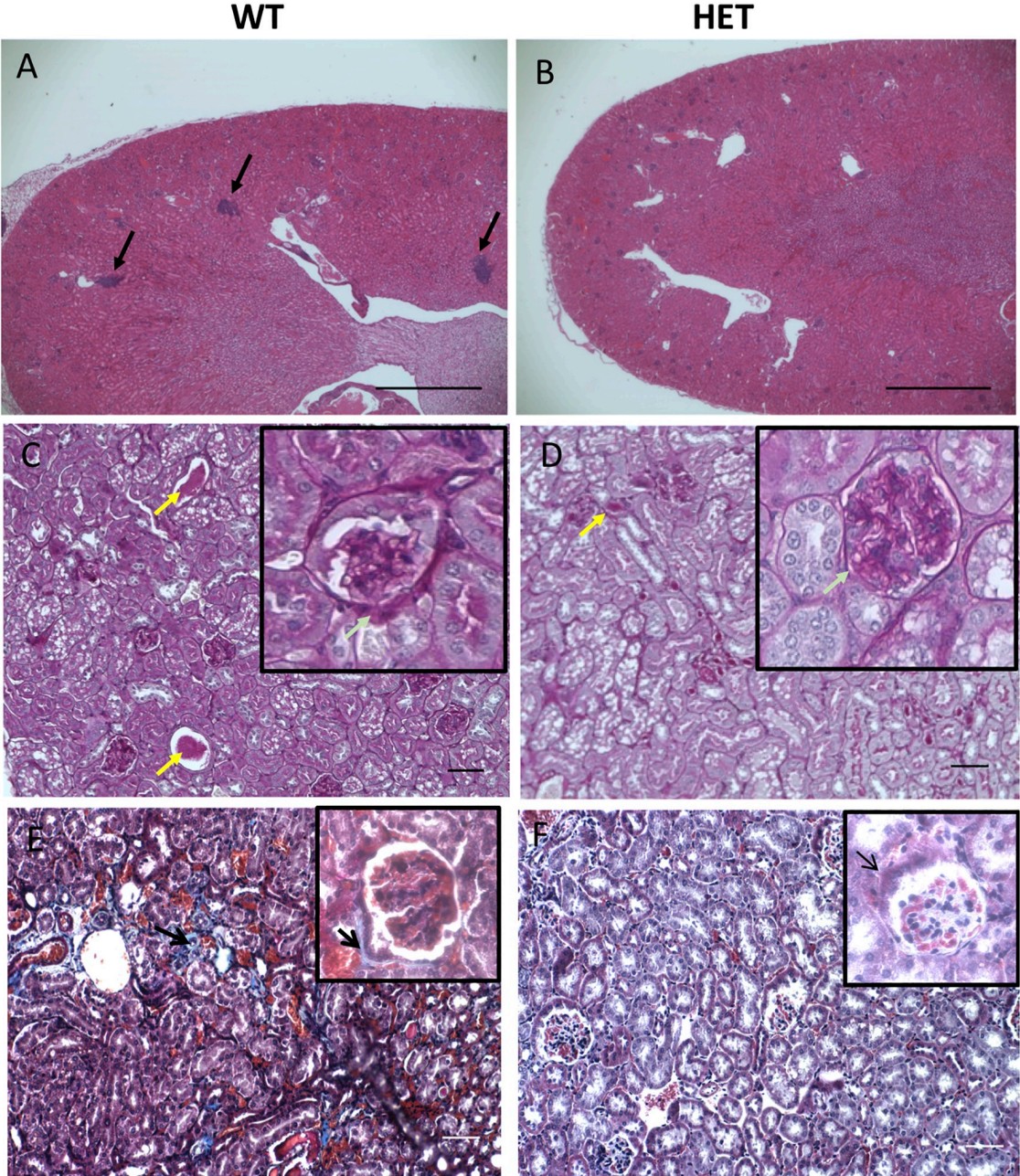

**Fig 2. Histological differences in representative renal sections from age-matched WT (n = 6) and HET (n = 6) mice at 18 months.** (A & B): Hematoxylin and Eosin staining in WT (A) and HET (B) kidneys showing evidence of increased inflammatory cell foci in WT (p = 0.0001) (Black arrows) as compared to HET animals. Original magnification 4X. Scale bar; 100 μm. (C & D): Periodic acid-Schiff (PAS) staining showing larger and more numerous intraluminal proteinaceous casts in WT (C) as compared to HET (D) mice (yellow arrow). Insert shows glomerulonephropathy (thickening of Bowman's capsule) in WT (green arrow) versus HET mice. (E & F): Masson-Trichrome staining of WT (E) and HET (F) kidney sections showing increased fibrosis indicated by the blue staining in WTs (Black arrows) as compared to HET mice. Original magnification for C, D, E and F: 20X. Scale bar; 50 μm.

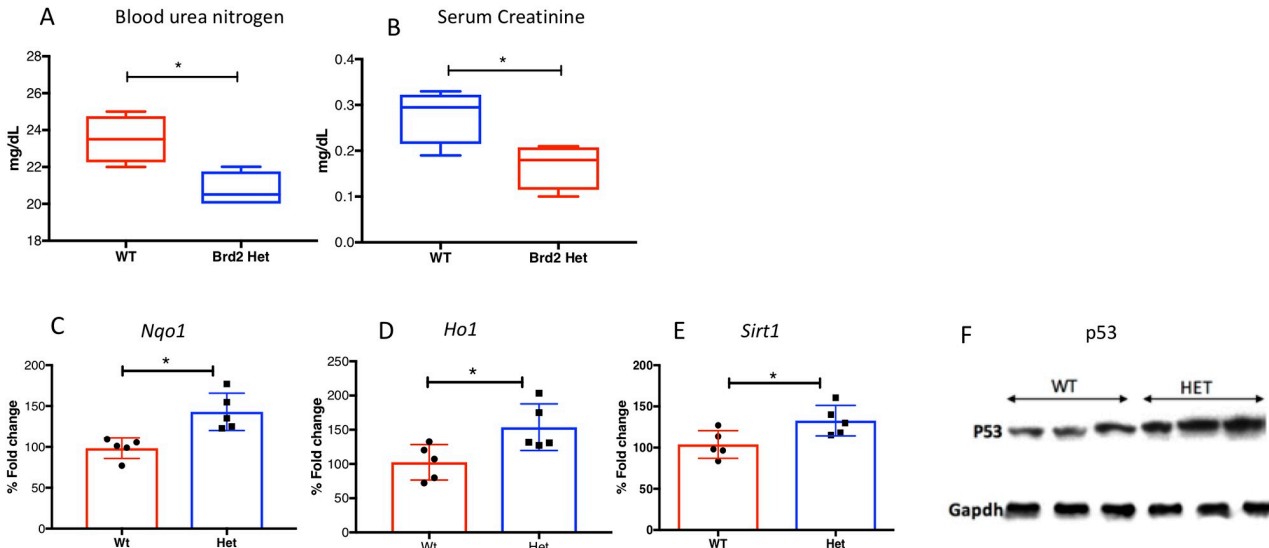

**Fig 3.** Biochemical and molecular changes in aged WT and HET mice: (A-B): Blood urea nitrogen (A) and serum creatinine levels (B) in WT and HET mice (n = 6/group) showing a significant increase in WT as compared to HET animals (p < 0.05). (C-E): Steady state levels *Nqo1*, *Hmox1* and *Sirt1* mRNA (n = 5/group) in WT and HET kidneys, showing upregulation of these cytoprotective genes in samples from HET mice (p <0.05). (F): Representative immunoblot showing elevated p53 levels in HET livers (n = 6/each group).

kidneys. Our *Brd2* haploinsufficient (HET) mice show significant lifespan extension and delays in age-related pathologies. Compared to WT mice, HETs displayed: (1) increased longevity (with or without tumors at necropsy), (2) reduced incidence of cancer at necropsy, (3) more youthful kidney structure and function, (4) increased expression of cytoprotective genes, and (5) a notable delay in age-related postural and behavioral phenotypes, (6) and extended fertility.

We hypothesize that healthspan and lifespan in humans could be increased by a reduction in BRD2 expression, in much the same way that our HETs show increased healthspan and lifespan relative to WTs. Support for this hypothesis follows from several observations. First, *Brd2* is overexpressed in several human tumors (TCGA database), and overexpression of *Brd2* (and other BET proteins) promotes cancer in mice [19, 21, 22]. Second, the persuit of cancer-related clinical trials for BET inhibitors [23–25, 35] suggests that Brd2 haploinsufficiency reduces cancer incidence in humans, just as our HET mice showed markedly reduced cancer incidence compared to WT (Table 3). Thus, inhibiting the activity of BRD2 in humans may increase longevity (and/or healthier aging) by reducing cancer incidence.

Interestingly, the decision to pursue *Cisd2* as a candidate longevity gene was motivated by a human family study of extremely old siblings [36]. Similarly, the decision to pursue BRD2 was motivated by a family study of human epilepsy. Given that *Cisd2* and *Brd2* appear to have a heritable influence on longevity in humans, and given that both genes increase lifespan in C57BJ/6 mice, *Cisd2* and *Brd2* should be considered among the top candidate genes for longevity.

It is important to note that, while three independent mouse models show reduced levels of Brd2, the Wang et al. [37] haploinsufficient model gives rise to obese mice—a phenotype not seen in our mice [3] nor in the mice of Gyuris et al. [26]. Furthermore, the Wang et al. (2009) [37] mice also show residual *Brd2* expression among BRD2$^{(-/-)}$ double knockout mice. By contrast, our mouse model [3]and the mouse model of Gyuris et al. (2009) [26] gave rise to the same phenotype, namely dysmorphic brain development in homozygous knockout mice. Both

models used the same gene trap (ES cell RRE050); our mice also showed seizure susceptibility and changes in brain structure in HETS [1, 2].

We did not observe appreciable difference in the median lifespans of our WT males and the Jackson Lab (JAX) C57BL/6J mouse colony. In our longevity tabulations we included *all* mice that died before adulthood, which Jackson labs does not. Furthermore, there is notable variations in average lifespan of this strain of mice across vivaria, especially differences in deciding when and with what pathologies aging mice should be euthanized [38, 39]. Furthermore, although the salient comparison is the one between our HETs and our WTs, it is also true that our HETs live longer (on average) than Jackson Labs C57BL/6J WT mice (p<0.002) [40].

## Molecular mechanisms for Brd2-related aging

As noted in Introduction, there are several speculated mechanisms for mammalian aging that have received considerable attention. However it is unlikely that any one of them (in isolation) explains Brd2's wide-range of effects on aging. Furthermore, Brd2's effect on aging is at a most basic level, as evidenced by the striking differences in biological ages between HET and WT age-matched mice as measured by the "methylation clock". As such, we believe that the *Brd2* HET mouse is an excellent place to start unraveling the combined effects of multiple age-related mechanisms while possibly uncovering novel mechanisms as well that could explain the differences we see in lifespan and healthspan.

## The significance of cancer reduction in Brd2 HET mice

We observed a lower incidence of tumors in HET mice, suggesting that these animals are protected from developing tumors. Furthermore, we find that tumor bearing HET mice live longer, not only than tumor-bearing WT mice, but also longer than WT mice without tumors. This suggests that either the onset of tumors begins later in HET mice, or the growth of tumors is slower in tumor bearing HET mice than in tumor bearing WT mice. In our earlier work, cell cycle analysis of mouse embryonic fibroblasts derived from double knockout embryos (*Brd2*$^{-/-}$, *i.e.*, embryos lacking all *Brd2*), resulted in cell cycle delay at the G1-S transition [3]. Thus, the apparent tumor-inhibition in the HET mice could be due to Brd2's function as an important cell cycle regulator. Because HETs live longer than WT mice whether cancer is present or not, cell cycle-related phenomena alone cannot explain the increase in longevity. Alternatively, many cytoprotective genes (e.g. *p53*, *Sirtuin1*, and *NRF2*) overlap with aging-related pathways that BRD2 is known to regulate [11, 20, 33]. These cytoprotective pathways include: IGF-Insulin, mTOR, p53, and DNA damage signaling. Therefore, we hypothesize that heightened expression of cytoprotective genes is crucial mechanism explaining Brd2-induced increases in healthspan and lifespan.

## Oxidative stress and reduced kidney pathology

Oxidative stress is a major contributor to kidney damage [41], and increased levels of BRD2 induce oxidative stress [33] in human cells. Therefore, *Brd2* haploinsufficiency, which reduces oxidative stress, could result in the healthier kidneys seen in HETs compared to WTs mice. This idea is supported by our results, and the results of others [33], showing upregulation of these antioxidants: *Nqo1* and *Hmox1* in HET kidneys. Our histopathological and kidney function results confirm this prediction. The youthful kidney anatomy seen in aged HET mice likely results, in part, from reduced renal oxidative stress.

## Longevity genes and kidney pathology

Some reno-protective genes are also reported to longevity-related (e.g. Klotho, PPARγ) [42, 43]. An example is *SIRT1* [12, 44–46], whose upregulation protects the kidney by reducing

inflammation [47, 48], and whose over-expression has also been associated with longevity [12]. We observed upregulation of *SIRT1* and reduced inflammation in the kidneys of our long-lived HET mice. These data suggest that *Brd2* haploinsufficiency promotes kidney youth by reducing inflammation through the upregulation of SIRT1 [11]. Crucially, Sirt1 is associated with anti-aging effects in hypothalamus [49], cortex, striatum [50–53], hippocampus [54, 55], as well as the kidney [56].

Chronic inflammation accelerates aging [57] and is one of the determinants of mortality [58]. It is interesting that BET proteins are critical for inflammatory response and Brd2 is essential for pro-inflammatory cytokine production [59]. Emerging preclinical and clinical evidence show a number of small molecule BET inhibitors with a potent anti-inflammatory activity [60, 61]. Thus, the reduced inflammation observed in HET kidneys could be due to Brd2's up-regulation of cytoprotective genes when it is haplodeficient, but it could also be due to a down-regulation of pro-inflammatory genes. Extreme longevity observed in our model could be the result of either the production of anti-inflammatory cytoprotective genes or a direct effect of on inflammatory genes, but in either case, caused by *Brd2* haploinsufficiency.

Taken together, our data implicate *Brd2* as a gene influencing longevity and healthspan. Brd2 haploinsufficiency significantly reduces the incidence of cancer at necropsy and facilitates healthier aging. The *decrease* in Brd2 is positively correlated with *increased* cytoprotective signaling through pathways like p53-, Sirt1- and Nrf2. These pathways regulate aging-related processes such as cellular senescence, apoptosis, genome maintenance, inflammation and cellular stress mitigation. Furthermore, since Brd2 has both genetic and epigenetic functions [62], it suggests that interconnected genetic, and likely epigenetic, mechanisms are involved in the regulation of longevity. We hypothesize that the extreme longevity seen in our HET mice is mediated by *Brd2* haploinsufficiency through the up-regulation of cytoprotective genes. Our results demonstrate—for the first time—a pronounced role of Brd2 in healthier murine aging, making Brd2 an excellent target for increasing both the lifespan and healthspan of mammals.

## Supporting information

**S1 Fig. qPCR analysis of tissue and MEFs from WT and *Brd2*+/- mice measuring the levels of *Brd2* transcript.**
(DOCX)

**S2 Fig. Although it is well known that caloric restriction is associated with increased lifespan in C57B6/J mice, we do not see any evidence in support of body weight differences in our HET and WT mice.**
(DOCX)

**S3 Fig. Histological differences in representative liver, spleen and testis sections from age matched WT and HET mice (n = 6) at 18 months.** (A & B): Hematoxylin & Eosin staining in WT liver (A) showing inflammation (thin black arrows) and more macrovascular vacuolation (thick, black arrow) in WT than HET (B). (C & D) Hematoxylin & Eosin staining in WT spleen (C) and HET(B) showing disorganization of the splenic structure in WT. (E & F) Hematoxylin & Eosin staining in WT (A) and HET(B) testes showing increased vacuolation in WT (C) as compared to HETs (D).
(DOCX)

**S1 Data.**
(XLSX)

**S1 Raw images.**
(DOCX)

## Acknowledgments

The authors thank Dr. Meng Wang for the statistical analysis of data. We would like to acknowledge Ms. Emily Cameron and Mr. Vraj Patel for assistance with data collection and genotyping.

## Author Contributions

**Conceptualization:** Shilpa Pathak, David A. Greenberg.

**Data curation:** Shilpa Pathak.

**Formal analysis:** Shilpa Pathak, William C. L. Stewart.

**Investigation:** Shilpa Pathak.

**Project administration:** Shilpa Pathak, David A. Greenberg.

**Supervision:** William C. L. Stewart, David A. Greenberg.

**Writing – original draft:** Shilpa Pathak, David A. Greenberg.

**Writing – review & editing:** Shilpa Pathak, William C. L. Stewart, Christin E. Burd, Mark E. Hester, David A. Greenberg.

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
