## [Decision Letter · Decision Letter 0]

16 Jan 2020

PONE-D-19-32190

Brd2 haploinsufficiency extends lifespan and healthspan in C57B6/J mice

PLOS ONE

Dear Dr Pathak,

Thank you for submitting your manuscript to PLOS ONE. After careful consideration, we feel that it has merit but does not fully meet PLOS ONE’s publication criteria as it currently stands. Therefore, we invite you to submit a revised version of the manuscript that addresses the points raised during the review process.

As you can see from the accompanying critiques the reviewers find merit in your work but request additional information before the manuscript can be considered for publication. This relates for example to the documentation of pathology in the animal cohorts. Also, please comment on point 1 raised by reviewer 2 regarding the lifespan of the control animals. Please also address all additional points in a revised version of your manuscript. 

We would appreciate receiving your revised manuscript by Feb 22 2020 11:59PM. To enhance the reproducibility of your results, we recommend that if applicable you deposit your laboratory protocols in protocols.io, where a protocol can be assigned its own identifier (DOI) such that it can be cited independently in the future. For instructions see: http://journals.plos.org/plosone/s/submission-guidelines#loc-laboratory-protocols

We look forward to receiving your revised manuscript.

Kind regards,

Christoph Englert

Academic Editor

PLOS ONE

Journal Requirements:

2. To comply with PLOS ONE submissions requirements, please provide method(s) of sacrifice in the Methods section of your manuscript.

3. As part of your revision, please complete and submit a copy of the ARRIVE Guidelines checklist, a document that aims to improve experimental reporting and reproducibility of animal studies for purposes of post-publication data analysis and reproducibility: https://www.nc3rs.org.uk/arrive-guidelines. Please include your completed checklist as a Supporting Information file. Note that if your paper is accepted for publication, this checklist will be published as part of your article.

4. Please compile all raw blot and gel images in a single PDF file titled S1_raw_images, and upload this file as Supporting Information or provide it via a publicly available data repository and include the dataset identifier (DOI or equivalent) in the Data Availability Statement. We ask that you ensure that every image in the file is clearly labeled to annotate the loading order, identity of experimental samples, method used to capture the image, and which figure panel was generated from that original image. Molecular weight markers should be included or indicated on each blot/gel image, and any lanes not included in the final figure should be marked with an “X” above the lane label on the original blot/gel image. All labeling and annotations should be performed without obscuring any data or background bands. Please note, there is a 20 MB maximum file size for Supporting Information files. If your PDF size is larger, please use a suitable repository or discuss with the journal staff.

5. Please note that all PLOS journals ask authors to adhere to our policies for sharing of data and materials: https://journals.plos.org/plosone/s/data-availability. According to PLOS ONE’s Data Availability policy, we require that the minimal dataset underlying results reported in the submission must be made immediately and freely available at the time of publication. As such, please remove any instances of 'unpublished data' or 'data not shown' in your manuscript and replace these with either the relevant data (in the form of additional figures, tables or descriptive text, as appropriate), a citation to where the data can be found, or remove altogether any statements supported by data not presented in the manuscript.

6. In your response letter, please note whether your blot/gel image data are in Supporting Information or posted at a public data repository, provide the repository URL if relevant, and provide specific details as to which raw blot/gel images, if any, are not available.

7. Thank you for your ethics statement: "All animal procedures were reviewed and approved under IACUC protocol (AR12-00016) at Nationwide Children's Hospital, Columbus, OH."

Please amend your current ethics statement to include the full name of the ethics committee that approved your specific study.

For additional information about PLOS ONE submissions requirements for ethics oversight of animal work, please refer to http://journals.plos.org/plosone/s/submission-guidelines#loc-animal-research 

8. In your Data Availability statement, you have not specified where the minimal data set underlying the results described in your manuscript can be found. PLOS defines a study's minimal data set as the underlying data used to reach the conclusions drawn in the manuscript and any additional data required to replicate the reported study findings in their entirety. All PLOS journals require that the minimal data set be made fully available. For more information about our data policy, please see http://journals.plos.org/plosone/s/data-availability.

9. PLOS ONE now requires that authors provide the original uncropped and unadjusted images underlying all blot or gel results reported in a submission’s figures or Supporting Information files. This policy and the journal’s other requirements for blot/gel reporting and figure preparation are described in detail at https://journals.plos.org/plosone/s/figures#loc-blot-and-gel-reporting-requirements and https://journals.plos.org/plosone/s/figures#loc-preparing-figures-from-image-files. When you submit your revised manuscript, please ensure that your figures adhere fully to these guidelines and provide the original underlying images for all blot or gel data reported in your submission. See the following link for instructions on providing the original image data: https://journals.plos.org/plosone/s/figures#loc-original-images-for-blots-and-gels.  

10. PLOS requires an ORCID iD for the corresponding author in Editorial Manager on papers submitted after December 6th, 2016. Please ensure that you have an ORCID iD and that it is validated in Editorial Manager. To do this, go to ‘Update my Information’ (in the upper left-hand corner of the main menu), and click on the Fetch/Validate link next to the ORCID field. This will take you to the ORCID site and allow you to create a new iD or authenticate a pre-existing iD in Editorial Manager. Please see the following video for instructions on linking an ORCID iD to your Editorial Manager account: https://www.youtube.com/watch?v=_xcclfuvtxQ

11. We note that you have included the phrase “data not shown” in your manuscript. Unfortunately, this does not meet our data sharing requirements. PLOS does not permit references to inaccessible data. We require that authors provide all relevant data within the paper, Supporting Information files, or in an acceptable, public repository. Please add a citation to support this phrase or upload the data that corresponds with these findings to a stable repository (such as Figshare or Dryad) and provide and URLs, DOIs, or accession numbers that may be used to access these data. Or, if the data are not a core part of the research being presented in your study, we ask that you remove the phrase that refers to these data.

12. Please include captions for your Supporting Information files at the end of your manuscript, and update any in-text citations to match accordingly. Please see our Supporting Information guidelines for more information: http://journals.plos.org/plosone/s/supporting-information.

Reviewers' comments:

Reviewer's Responses to Questions

**Comments to the Author**

1. Is the manuscript technically sound, and do the data support the conclusions?

Reviewer #1: Yes

Reviewer #2: Partly

2. Has the statistical analysis been performed appropriately and rigorously? 

Reviewer #1: Yes

Reviewer #2: Yes

3. Have the authors made all data underlying the findings in their manuscript fully available?

Reviewer #1: Yes

Reviewer #2: Yes

4. Is the manuscript presented in an intelligible fashion and written in standard English?

Reviewer #1: Yes

Reviewer #2: Yes

5. Review Comments to the Author

Reviewer #1: In this manuscript, the authors observed that reduced Brd2 expression benefits for extending lifespan and healthspan of C57B6/J mice. Despite Brd2 haploinsufficient mice have been generated by several groups in different ways, the pro-longevity phenotype in HET mice has never been identified. The authors also concerned the downstream effectors, e.g. Sirt1, HO-1, and P53, were upregulated in Brd2 HET tissues. These genes are apparently cytoprotective and longevity-related, partially explaining the pronounced role of Brd2 haploinsufficiency in longevity. However, there are a number of issues the authors need to address before publication.

In the second paragraph, the authors introduced Cisd2, a well-known longevity gene, prolongs the lifespan of mice to the extent that Brd2 haploinsufficiency does. Does the authors aim to exclude the benefits of Brd2 HET from Cisd2’s role? Are there some functional overlaps between these two proteins? However, I could not find any test on Cisd2 function in Brd2 HET mice.

Since the authors’ observation is so distinct to the other groups based on their mouse models, please provide the generating strategy and characterization of the Brd2 HET mice, e.g. scheme of gene trap and Brd2 protein expression profile in various tissues.

Again please discuss a bit more the reason why your mouse models are phenotypically different to other groups.

In supplementary Fig2, the author showed the bodyweight was unchanged in Brd2 HET mice when comparing to WT. However, it has been reported that Brd2 disruption in mice causes severe obesity. Please discus this difference.

In addition to kidney, the authors also definitely examined the pathologies in other organs, like liver, spleen and testis (data not shown). I suggest the authors provide these data to strengthen the conclusion that Brd2 HET improves mouse healthspan.

In this manuscript, the authors provided large descriptive information on mouse aging phenotype, yet no evidence. For example, in line 262-271, the whole paragraph demonstrated the observation of symptoms of aging in WT and HET mice without any data. The authors should provide some scientific evidence, like the pictures of aged WT and HET mice, quantifications of mouse activities, or the movies on mouse behavior …

Please label the medium lifespan of mice in figure 1. In Result section, there is no data description of Figure 2C.

Reviewer #2: This paper describes a novel discovery, that reduced expression of Brd2 leads to increased lifespan, which is associated with reduced pathology and improved health. Although this is an important observation, I have the following major concerns with the manuscript:

1. The increase in lifespan is impressive and is the key to this paper. However, the lifespan reported for the WT female and male mice (which are in the C57BL/6 background) of 23 and 26 months is very short compared to other reports (e.g., see data from Jackson Laboratory, de Cabo, and Richardson) for C57BL/6 mice. The lifespans the authors obtain with the Brd2 HET mice are actually about what is observed with normal WT mice. This is a major concern because the increased lifespan could arise from making the mice more robust to the conditions that gave rise to the shorter lifespan of the WT mice rather than retarding aging. This has been shown in previous studies, which reported increased lifespan when the median/mean lifespan of the WT mice was less thatn 26 months of age, and the increased lifespan was not replicated when conducted in colonies where the WT mice had a median/mean lifespan of over 29 months.

2. A great deal of emphasis was placed on the pathological data. The major cause of death in C57BL/6 mice is lymphoma in most aging colonies; however, the data on cancer was lumped together as tumors, i.e., it was not clear what the neoplastic lesions were in the mice. In addition, renal pathology appears to be a major problem in the mice in this study, which is in contrast to most other studies on aging in C57BL/6 mice, where renal pathology is relatively minor.

3. The data on the mice being ‘biologically’ younger was weak. For example, a great deal of emphasis was placed on the epigenetic clock. However, the epigenetic clock is a measure of chronological age, not physiological age. In addition, these data were generated with only 4 mice per group and no data were given of the animal to animal variation in data (e.g., SD or SEM). The data on observational health of the animals on page 13 was unconvincing. Data on physiological functions such as grip strength, activity, rotarod performance, and cognition would have been stronger evidence that the HET mice were physiologically younger.

Minor Concerns:

1. The authors state on page 4 (line 78-77) that there are “only a handful of candidate longevity genes for mice.” This is not correct; there have been at least two dozen different genes identified. The most cited are those that show reduced growth hormone/IGF. The Cisd2 mouse has not been studied greatly from an aging context.

2. In the methods, it was stated that the mice were either allowed to live out their lifespan or euthanized. It would be important to know the number (%) of mice euthanized for the WT and HET mice, i.e., where more WT mice euthanized.

3. Supplement: there is no list of references for the figures in the supplement, and it is not clear what the third figure is all about.

4. The introduction reads more like a discussion. For example, I found very little information about the Brd2 gene in the introduction or anywhere in the manuscript. I would have liked to know what protein this gene codes for and its biochemical/molecular function, etc.

6. PLOS authors have the option to publish the peer review history of their article (what does this mean?). If published, this will include your full peer review and any attached files.

Reviewer #1: No

Reviewer #2: No

---

## [Author Response · Author response to Decision Letter 0]

8 May 2020

Reviewer # 1 

In this manuscript, the authors observed that reduced Brd2 expression benefits for extending lifespan and healthspan of C57B6/J mice. Despite Brd2 haploinsufficient mice have been generated by several groups in different ways, the pro-longevity phenotype in HET mice has never been identified. The authors also concerned the downstream effectors, e.g. Sirt1, HO-1, and P53, were upregulated in Brd2 HET tissues. These genes are apparently cytoprotective and longevity-related, partially explaining the pronounced role of Brd2 haploinsufficiency in longevity. However, there are a number of issues the authors need to address before publication.

Comment 1: In the second paragraph, the authors introduced Cisd2, a well-known longevity gene, prolongs the lifespan of mice to the extent that Brd2 haploinsufficiency does. Does the authors aim to exclude the benefits of Brd2 HET from Cisd2’s role? Are there some functional overlaps between these two proteins? However, I could not find any test on Cisd2 function in Brd2 HET mice.

Response 1: First, our HET mice are identical to C57B6/J at every gene except Brd2, having been back-crossed for at least 10 generations. Thus, there is little or no probability that any meaningful Cisd2 expression differences between WTs and HETs exist. Second, there are no known functional overlaps between Cisd2 (which is chiefly involved in Calcium homeostasis) and Brd2 (which is an epigenetic reader). These points have now been included in the second paragraph of Introduction and Discussion (lines 430-435).

Comment 2: Since the authors’ observation is so distinct to the other groups based on their mouse models, please provide the generating strategy and characterization of the Brd2 HET mice, e.g. scheme of gene trap and Brd2 protein expression profile in various tissues.

Response 2: Respectfully, we must disagree with the Reviewer’s comment that our results are “so distinct” from other groups. First, Gyruis et al. (2009) (1) used the same ES cell line as we did (RRE050) and confirmed our findings (Shang et al. 2009) (2), including that Brd2-/- mice have impaired CNS development, an observation about which the Wang et al. (2009) (3) paper is silent. Second, Wang et al. used a Brd2 insertion/disrupter (e.g. ES cell line RRT234) different from the one we used or that Gyruis et al used. Third, Wang et al’s principal phenotype was obesity in the mice but neither Gyruis et al nor we observed obesity in our mice. Fourth, and most quizzically, there is residual Brd2 expression in their double knockout mouse, whereas neither Gyruis nor we observed any residual expression. Thus, the phenotype of Gyruis’s mouse and our mouse phenotype are not “distinct” whereas the Wang et al. phenotype differs markedly from our findings and those of Gyruis.

As to generating strategy, the genetic engineering and characteristics of Brd2 HET mice has been described in details in Shang et al (2009) (2), and we have added the mouse generation strategy in detail in the revised manuscript (see Materials and Methods, lines 143-151). 

Gene expression profiles showing (among other things) the reduction of Brd2 expression in HET animals in kidney, liver, heart and mouse embryonic fibroblasts is a part of supplemental information (Supplementary Figure 1). These findings were also replicated by Gyruis et al. (2009)(1).

Comment 3: Again, please discuss a bit more the reason why your mouse models are phenotypically different to other groups.

Response 3: This comment was addressed above in Response 2 and we have modified our discussion incorporating the genotypic and phenotypic differences related to the different mouse models (Discussion, lines 437-444).

Comment 4: In supplementary Fig2, the author showed the bodyweight was unchanged in Brd2 HET mice when comparing to WT. However, it has been reported that Brd2 disruption in mice causes severe obesity. Please discus this difference.

Response 4: Again, this comment was addressed in Response 2, and in Discussion.

Comment 5: In addition to kidney, the authors also definitely examined the pathologies in other organs, like liver, spleen and testis (data not shown). I suggest the authors provide these data to strengthen the conclusion that Brd2 HET improves mouse health span.

Response 5: The histopathology data for liver, spleen and testis has been included in the supplementary information in the revised manuscript.

Comment 6: In this manuscript, the authors provided large descriptive information on mouse aging phenotype, yet no evidence.. For example, in line 262-271, the whole paragraph demonstrated the observation of symptoms of aging in WT and HET mice without any data. The authors should provide some scientific evidence, like the pictures of aged WT and HET mice, quantifications of mouse activities, or the movies on mouse behavior …

Response 6: We are somewhat surprised by the comment about the acceptability our description of the differences over time between the HET and WT mice. Such descriptive content has long been a staple of scientific publications going back to the very beginning of science journals. Not surprisingly, standard photographs are often inadequate to capture the differences (e.g. subtle differences in grooming and grizzled appearance). Furthermore, while lifetime surveillance of each mouse would permit quantification of differences in maintenance of youthful activity versus continued decline in actively resulting in lethargy, this is impractical. Therefore, the time-honored practice of reporting such observations in the scientific literature are crucial to the understanding of phenotype.

As to mouse behavior, those results were published in Chachua et al. 2014 (4). Those studies showed that while HETs and WTs do have similar cognitive functioning, the HETs have decreased anxiety, and are more aggressive than WTs (especially in the case of females). 

Comment 7: Please label the medium lifespan of mice in figure 1. In Result section, there is no data description of Figure 2C. 

Response 7: The median lifespan has been labeled in Figure 1, and the Figure 2C has been mentioned in the revised manuscript (Line 350). 

Reviewer # 2 

This paper describes a novel discovery, that reduced expression of Brd2 leads to increased lifespan, which is associated with reduced pathology and improved health. Although this is an important observation, I have the following major concerns with the manuscript:

Comment 1A: The increase in lifespan is impressive and is the key to this paper. However, the lifespan reported for the WT female and male mice (which are in the C57BL/6 background) of 23 and 26 months is very short compared to other reports (e.g., see data from Jackson Laboratory, de Cabo, and Richardson) for C57BL/6 mice. 

Response 1A: In fact, there is no appreciable difference in the median lifespans of our WT males and the Jackson Lab (JAX) C57BL/6J mouse colony. In our longevity tabulations we included all mice that died before adulthood, which Jackson labs does not. Furthermore, there is notable variations in average lifespan of this strain of mice across vivaria, especially differences in deciding when and with what pathologies aging mice should be euthanized (5, 6). Furthermore, although the salient comparison is the one between our HETs and our WTs, it is also true that our HETs live longer (on average) than Jackson Labs C57BL/6J WT mice (p<0.002) (7). This is included in Discussion section in the revised manuscript. (See also the answer to 1C, below.)

Comment 1B: The lifespans the authors obtain with the Brd2 HET mice are actually about what is observed with normal WT mice. 

Response 1B: We think the reviewer is mistaken on this point. 31.5 months (HET median) is 23% longer than 26.5 months (WT median; p<0.0001). 

Comment 1C: This is a major concern because the increased lifespan could arise from making the mice more robust to the conditions that gave rise to the shorter lifespan of the WT mice rather than retarding aging. This has been shown in previous studies, which reported increased lifespan when the median/mean lifespan of the WT mice was less than 26 months of age, and the increased lifespan was not replicated when conducted in colonies where the WT mice had a median/mean lifespan of over 29 months. 

Response 1C: There are several reports in the literature of median lifespan for WT approx. 26-27 months, so by comparison, our HET animal with median lifespan of 31-32 months is still quite remarkable. The HET males showed a 19% extension in median lifespan (925 vs 781 days; p<0.0001) but the WT females appear to have shorter lives than the WT males: WT females show a median lifespan of 23 months, and we agree it would be interesting to know how Brd2 haploinsufficiency might compensate for it compared to 26 months for WT males, but we hypothesize that the reason for the difference is not genetic. The median lifespan of our WT males is the same as reported by other vivaria, but the mean figure for WT females is lower than the median lifespan of JAX C57. We hypothesize that the hyper-aggressiveness of the HET females (4), which are more aggressive than even the WT males, leads to increased stress on the WT females, which are housed with the HET females, and that this accounts for the somewhat shorter lifespan of the WT females, which also reduces the overall WT survival rate. We plan to test this hypothesis but currently, such investigations are beyond the scope of the current paper. Moreover, it is highly unlikely that the shortened lifespan of WT females is genetic in origin since our HETs (which were backcrossed for 10 generations with WTs) do not have hidden mutations or deletions (as determined by whole genome sequencing). 

Comment 2: A great deal of emphasis was placed on the pathological data. The major cause of death in C57BL/6 mice is lymphoma in most aging colonies; however, the data on cancer was lumped together as tumors, i.e., it was not clear what the neoplastic lesions were in the mice. In addition, renal pathology appears to be a major problem in the mice in this study, which is in contrast to most other studies on aging in C57BL/6 mice, where renal pathology is relatively minor.

Response 2: Our lab has considerable expertise in the area of kidney and liver pathology, and there is strong evidence in the aging literature (8, 9) that dysmorphic kidney and liver structure (e.g. lesions) is a normal part of the aging process in mice. Therefore, differences in the organ pathology of HET and WT mice are a natural place to start: both from the perspective of documenting what was observed, and from the perspective of generating mechanistic hypotheses to test. For instance, given the abnormal structure of both liver and kidney in WT (Supplementary Figure 3), it is possible that HET animals may have fewer senescent cells than WT. This would be an interesting hypothesis to test in future studies. The overwhelming point, however, is that the HETs show considerably less age-related organ pathology than the WT.

While lymphoma is most certainly an important cause of death in C57 (somewhere between 10% and 50% (10), our primary endpoint for this study is lifespan, not cancer. It was only incidentally (ie, at the time of death), that we noticed a difference in HETs and WTs with respect to the presence of solid tumor(s). While determination of the primary tumor type in these animals would be interesting, this is beyond the scope of this current work.

Comment 3A: The data on the mice being ‘biologically’ younger was weak. For example, a great deal of emphasis was placed on the epigenetic clock. However, the epigenetic clock is a measure of chronological age, not physiological age. 

Response 3A: The methylation clock (DNAge) was first demonstrated to be highly correlated with chronological age, and could then be used as a good predictor of biological age (11-15). 

Furthermore, as we illustrate in Table 1, WT mice of the same chronological age as HET mice (i.e., 770 days) have dramatically higher biologically measured age (DNAges) than the longer-lived HET mice. Since we know that HETs live longer than WTs (in general), this confirms several reports in the aging literature that DNAge is a better measure of biological age than is simple chronological age. 

Comment 3B: In addition, these data were generated with only 4 mice per group and no data were given of the animal to animal variation in data (e.g., SD or SEM). 

Response 3B: Even though there are only 4 mice per group, the methylation clock analysis showed statistically significant younger liver tissue in HET’s than WT’s (p<0.012 and SEM(WT): 4.5 and SEM(HET): 7.1), and the DNAges of each animal are shown in Table 2. We have included SEMs in the revised manuscript. 

Comment 3C: The data on observational health of the animals on page 13 was unconvincing. 

Response 3C: Please see Response 6 to Reviewer 1. 

Comment 3D: Data on physiological functions such as grip strength, activity, rotarod performance, and cognition would have been stronger evidence that the HET mice were physiologically younger.

Response 3D: We have previously tested HET and WT mice in a battery of behavioral tests (open field, tube dominance test, elevated plus maze, Morris water maze and Barnes maze) that showed increased aggressiveness in HETs without cognitive impairment (4). Relative to the biological indicators of age-related differences that we already have described (e.g. HETs have increased lifespan, HETs have reduced cancer incidence, and HET have reduced organ pathology), tracking the rate of cognitive decline will not give us a better indication of the HET-WT aging difference. Testing such physiological functions in HET mice is currently beyond the scope of this manuscript, and is contingent upon future funding.

Minor Concerns:

Comment 4: The authors state on page 4 (line 78-77) that there are “only a handful of candidate longevity genes for mice.” This is not correct; there have been at least two dozen different genes identified. The most cited are those that show reduced growth hormone/IGF. The Cisd2 mouse has not been studied greatly from an aging context.

Response 4: By our count, we find that there are only 11 candidate longevity genes for C57 mice (16), and we now express this explicitly in the revised manuscript in the second paragraph of Introduction. Moreover, of the aforementioned 11 candidates, only one has greater median lifespan than our HETs, and two actually have reduced lifespan compared to JAX labs.

Comment 5: In the methods, it was stated that the mice were either allowed to live out their lifespan or euthanized. It would be important to know the number (%) of mice euthanized for the WT and HET mice, i.e., where more WT mice euthanized. 

Response 5: In the WT cohort, 68% mice were euthanized as opposed to 45% mice in HET group. 

Comment 6: Supplement: there is no list of references for the figures in the supplement, and it is not clear what the third figure is all about. 

Response 6: Figures in the Supplementary information has been appropriately cited in the manuscript and the third figure in the supplementary information has been deleted. 

Comment 7: The introduction reads more like a discussion. For example, I found very little information about the Brd2 gene in the introduction or anywhere in the manuscript. I would have liked to know what protein this gene codes for and its biochemical/molecular function, etc. 

Response 7: The reviewer raises an important stylistic point. We did not include much information about Brd2’s biochemical/molecular function because, as interesting as that would be from a basic science point of view, we have yet to find a plausible explanation for the wide range of biological processes in which this molecule is involved. Aside from its two most cited roles: a transcription factor element in copying RNA from DNA, and an acetylated histone effector, it is associated with a number of disparate biologic observations, such as: over-expression in cancer; regulation of neuron migration during development; linkage and association to a specific form of epilepsy; over-expression in hormonally modulated epithelia (e.g., mammary gland, ovary, kidney, and uterus); and it is even reported to act as a scaffold for TATA binding. Thus, in our view, what the protein “does” is not at all well understood. Furthermore, because it takes a great deal of space to explain what individual processes the protein is involved in, and because this explanation is unlikely to provide an overarching understanding of how Brd2 brings about these observations, we did not feel it at all appropriate to go into detail about topics that do not, as far as we can see, increase understanding of Brd2’s effect on mouse lifespan and health. Instead, we opted for explaining only how we came to serendipitously observe its effect on longevity and to substantiate the depth of that effect with observations, for example, on changes in kidney pathology between HETs and WTs. In truth, we regret not being able to review Brd2’s biologic functions in greater detail than the few lines allotted in Discussion, but that task is best left to a review article, not in this report—which emphasizes Brd2’s dramatic and surprisingly beneficial effect on mouse longevity and health.

References:

1. Gyuris A, Donovan DJ, Seymour KA, Lovasco LA, Smilowitz NR, Halperin AL, et al. The chromatin-targeting protein Brd2 is required for neural tube closure and embryogenesis. Biochim Biophys Acta. 2009;1789(5):413-21.

2. Shang E, Wang X, Wen D, Greenberg DA, Wolgemuth DJ. Double bromodomain-containing gene Brd2 is essential for embryonic development in mouse. Dev Dyn. 2009;238(4):908-17.

3. Wang F, Liu H, Blanton WP, Belkina A, Lebrasseur NK, Denis GV. Brd2 disruption in mice causes severe obesity without Type 2 diabetes. Biochem J. 2009;425(1):71-83.

4. Chachua T, Goletiani C, Maglakelidze G, Sidyelyeva G, Daniel M, Morris E, et al. Sex-specific behavioral traits in the Brd2 mouse model of juvenile myoclonic epilepsy. Genes Brain Behav. 2014;13(7):702-12.

5. Yuan R, Peters LL, Paigen B. Mice as a mammalian model for research on the genetics of aging. ILAR J. 2011;52(1):4-15.

6. Yuan R, Tsaih SW, Petkova SB, Marin de Evsikova C, Xing S, Marion MA, et al. Aging in inbred strains of mice: study design and interim report on median lifespans and circulating IGF1 levels. Aging Cell. 2009;8(3):277-87.

7. Flurkey K. The Mouse in Biomedical Research. 2nd Edition ed. Burlington, MA: Elsevier; 2007.

8. Lim JH, Kim EN, Kim MY, Chung S, Shin SJ, Kim HW, et al. Age-associated molecular changes in the kidney in aged mice. Oxid Med Cell Longev. 2012;2012:171383.

9. Pettan-Brewer C, Treuting PM. Practical pathology of aging mice. Pathobiol Aging Age Relat Dis. 2011;1.

10. Ward JM. Lymphomas and leukemias in mice. Exp Toxicol Pathol. 2006;57(5-6):377-81.

11. Bocklandt S, Lin W, Sehl ME, Sanchez FJ, Sinsheimer JS, Horvath S, et al. Epigenetic predictor of age. PLoS One. 2011;6(6):e14821.

12. Zhang Y, Wilson R, Heiss J, Breitling LP, Saum KU, Schottker B, et al. DNA methylation signatures in peripheral blood strongly predict all-cause mortality. Nat Commun. 2017;8:14617.

13. Petkovich DA, Podolskiy DI, Lobanov AV, Lee SG, Miller RA, Gladyshev VN. Using DNA Methylation Profiling to Evaluate Biological Age and Longevity Interventions. Cell Metab. 2017;25(4):954-60 e6.

14. Field AE, Robertson NA, Wang T, Havas A, Ideker T, Adams PD. DNA Methylation Clocks in Aging: Categories, Causes, and Consequences. Mol Cell. 2018;71(6):882-95.

15. Horvath S. DNA methylation age of human tissues and cell types. Genome Biol. 2013;14(10):R115.

16. Pedro de Magalhaes J, Thompson L, de Lima I, Gaskill D, Li X, Thornton D, et al. A Reassessment of Genes Modulating Aging in Mice Using Demographic Measurements of the Rate of Aging. Genetics. 2018;208(4):1617-30.

---

## [Decision Letter · Decision Letter 1]

5 Jun 2020

Brd2 haploinsufficiency extends lifespan and healthspan in C57B6/J mice

PONE-D-19-32190R1

Dear Dr. Pathak,

We’re pleased to inform you that your manuscript has been judged scientifically suitable for publication and will be formally accepted for publication once it meets all outstanding technical requirements.

Kind regards,

Christoph Englert

Academic Editor

PLOS ONE

Additional Editor Comments (optional):

Reviewers' comments:

Reviewer's Responses to Questions

**Comments to the Author**

1. If the authors have adequately addressed your comments raised in a previous round of review and you feel that this manuscript is now acceptable for publication, you may indicate that here to bypass the “Comments to the Author” section, enter your conflict of interest statement in the “Confidential to Editor” section, and submit your "Accept" recommendation.

Reviewer #1: All comments have been addressed

2. Is the manuscript technically sound, and do the data support the conclusions?

Reviewer #1: (No Response)

3. Has the statistical analysis been performed appropriately and rigorously? 

Reviewer #1: (No Response)

4. Have the authors made all data underlying the findings in their manuscript fully available?

Reviewer #1: (No Response)

5. Is the manuscript presented in an intelligible fashion and written in standard English?

Reviewer #1: (No Response)

6. Review Comments to the Author

Reviewer #1: (No Response)

7. PLOS authors have the option to publish the peer review history of their article (what does this mean?). If published, this will include your full peer review and any attached files.

Reviewer #1: No

---

## [Editor Report · Acceptance letter]

10 Jun 2020

PONE-D-19-32190R1 

*Brd2* haploinsufficiency extends lifespan and healthspan in C57B6/J mice 

Dear Dr. Pathak:

I'm pleased to inform you that your manuscript has been deemed suitable for publication in PLOS ONE. Congratulations! Your manuscript is now with our production department. 

Kind regards, 

on behalf of

Dr. Christoph Englert 

Academic Editor

PLOS ONE